# Enhancing lobaplatin sensitivity in lung adenocarcinoma through inhibiting LDHA-targeted metabolic pathways

Siyu Yuan[1‡], Wenjie Ou[2‡], Xuguang Mi[3], Junjie Hou[4]*

1 Department of Clinical Nutrition, The Second Affiliated Hospital, Zhejiang University School of Medicine, Hangzhou, Zhejiang, China, 2 School of Clinical Medicine, Changchun University of Chinese Medicine, Changchun, Jilin, China, 3 Department of central laboratory, Jilin Provincial People's Hospital, Changchun, Jilin, China, 4 Department of Comprehensive Oncology, Jilin Provincial People's Hospital, Changchun, Jilin, China

‡ SY and WO are contributed equally to this work and should be considered co-first authors.
* houjunjie1979@163.com

**Data Availability Statement:** All relevant data are within the manuscript and its Supporting Information files.

**Funding:** Financial support was provided by Jilin Province Health Commission Project (2021LC058),

## Abstract

### Background and objective

Lung adenocarcinoma (LUAD), a subtype of non-small cell lung cancer (NSCLC), is associated with high incidence and mortality rates. Effective treatment options are limited due to the frequent development of multidrug resistance, making it crucial to identify new therapeutic targets and sensitizing agents. This study investigates the role of Lactate dehydrogenase A (LDHA) in enhancing the chemotherapy sensitivity of Lobaplatin (LBP) in LUAD.

### Methods

Bioinformatics analyses were performed using data from The Cancer Genome Atlas (TCGA) and Gene Expression Omnibus (GEO) to assess LDHA expression in LUAD tissues. LUAD cell lines A549 and NCL-H1975 were treated with siRNA targeting LDHA and the small molecule inhibitor Oxamate. We measured changes in lactate production, ATP levels, NAD+ and pyruvate levels, and assessed cell viability. The chemotherapy sensitivity to Lobaplatin was evaluated, and key signaling pathways related to chemotherapy resistance were analyzed.

### Results

The inhibition of LDHA resulted in a significant reduction in lactate production and ATP levels, along with an increase in NAD+ and pyruvate levels. These metabolic alterations led to decreased cell viability and enhanced sensitivity to Lobaplatin. The study identified the PI3K/AKT signaling pathway as a critical mediator of this enhanced sensitivity, with reduced phosphorylation of AKT observed upon LDHA inhibition. Furthermore, the combination of LDHA inhibition and Lobaplatin treatment demonstrated a synergistic effect, significantly inhibiting tumor growth and highlighting the potential of LDHA as a therapeutic target to overcome drug resistance in LUAD.

Jilin Province Science and Technology Development Plan Project (20230402007GH), and Jilin Province Health Science and Technology Ability Improvement Project (2023LC043). The funders had no role in study design, data collection and analysis, decision to publish, or preparation of the manuscript.

**Competing interests:** The authors have declared that no competing interests exist.

**Abbreviations:** GEO, Gene Expression Omnibus; KEGG, Kyoto Encyclopedia of genes and Genomes; LDHA, Lactate dehydrogenase A; LBP, Lobaplatin; LUAD, Lung Adenocarcinoma; NSCLC, Non-small cell lung cancer; TCGA, The Cancer Genome Atlas.

## Conclusion

Targeting LDHA and disrupting lactate metabolism and its signaling pathways can effectively enhance the sensitivity of LUAD to Lobaplatin, providing a promising approach to overcoming multidrug resistance. These findings offer valuable insights into developing new treatment strategies for lung adenocarcinoma, emphasizing the role of metabolic pathways in cancer therapy.

## 1. Introduction

The incidence and mortality rates of lung cancer in China have significantly risen, with predictions indicating a continual increase [1]. Non-small cell lung cancer (NSCLC) accounts for approximately 85% of lung cancer cases, with Lung Adenocarcinoma (LUAD) constituting about 40% [2]. Lobaplatin (LBP), a third-generation platinum-based antitumor drug, exhibits superior treatment metrics and low toxicity, without cross-resistance to cisplatin [3]. Several clinical trials and published studies have shown LBP's antitumor activity in various cancer types, including breast cancer [4], colorectal cancer [5], hepatocellular carcinoma [6], small cell lung cancer [7], and melanoma [8], etc. Platinum-based dual-drug regimens are now standard first-line treatments for NSCLC. However, tumor resistance to LBP treatment is inevitable. Exploring novel drugs in combination with standardized chemotherapy is a crucial strategy to overcome resistance.

Otto Warburg first described in 1924 that cancer cells exhibit higher glucose uptake and rely more on aerobic glycolysis for ATP production, known as the "Warburg effect" [9]. As cancer cells increase energy production for cell growth and to sustain proliferation under hypoxic conditions, elevated glycolytic activity ensures sufficient ATP levels [10]. Recently, targeting energy metabolism has returned to the forefront of cancer treatment, with more details and molecular mechanisms involved in the "Warburg effect" being discovered [11]. This not only improves our understanding of cancer cell characteristics but also provides a molecular mechanism basis for killing cancer cells. Among the numerous enzymes involved in glycolysis, LDHA, a key enzyme in aerobic glycolysis, catalyzes the mutual conversion of lactate and pyruvate, along with the interconversion of NADH and NAD+ [12], making it a promising anti-cancer target [11].

In summary, LDHA may represent a promising therapeutic target for enhancing LBP's chemotherapy sensitivity in LUAD. This study aims to verify LDHA's expression and metabolic impact in human LUAD and understand the role of LDHA in tumor treatment and drug sensitivity in LUAD cells.

## 2. Materials and methods

### 2.1 Bioinformatics analysis

Transcriptome data for 516 cases of LUAD tissue and 59 cases of normal tissue, along with clinical data in TPM format, were downloaded from The Cancer Genome Atlas (TCGA, https://cancergenome.nih.gov/) database. Gene array expression data GSE68571 and its platform file GPL80-30376 were obtained from the Gene Expression Omnibus (GEO, https://www.ncbi.nlm.nih.gov/geo/) database. The "Limma" R package in R software (R 4.3.1, http://www.R-project.org/) was used for differential analysis of LDHA expression, and the "survival"

package in R was utilized to draw Kaplan-Meier survival curves to estimate if there was a significant difference in survival times.

## 2.2 Gene enrichment analysis

Based on the median value of LDHA expression, the TCGA dataset was divided into high LDHA expression group and low LDHA expression group. Kyoto Encyclopedia of Genes and Genomes (KEGG) pathway enrichment analysis was conducted to further understand the disease phenotypes and biological processes associated with LDHA expression.

## 2.3 Patient source

Ethical approval for procedures involving human subjects was granted by the Ethics Committee of Jilin Province People's Hospital (Approval number: 201902). Three cases of LUAD tissue and paired adjacent non-tumor tissue were obtained from patients who underwent surgery with rapid pathological confirmation at Jilin Province People's Hospital between 2022 and 2023. Written informed consent was obtained from all patients.

## 2.4 Cell lines

Human LUAD cell lines A549 and NCL-H1795 were preserved by the central laboratory of Jilin Province People's Hospital. Cells were cultured in Dulbecco's Modified Eagle's Medium (DMEM) high glucose medium supplemented with 10% fetal bovine serum (FBS, Trans, FS301, China) and 1% penicillin-streptomycin solution (MCE, HY-K1006, USA) at 37°C, 100% humidity, and 5% CO2. All cell lines were maintained as monolayer cultures, and upon reaching 70–90% confluence, cells were passaged and the medium was refreshed. The passage range for all cell lines was between 1 and 25. All cell lines were tested and found to be mycoplasma-free. All cells were authenticated before use.

## 2.5 Plasmid transfection

Three siRNA sequences targeting LDHA were designed, with the most effective siRNA sequence used for subsequent transfections. The sense strand sequences targeting LDHA are as follows: LDHA siRNA-1 (Si-1) 5′-GGACTTGGCAGATGAACTTGC-3′; LDHA siRNA-2 (Si-2) 5′-CCAAAGATTGTCTCTGGCAAA-3′; LDHA siRNA-3 (Si-3) 5′-GGTACCACTTCCATTGTAAGT-3′. The sequence for the negative control siRNA was: 5′-CCTAAGGTTAAGTCGCCCTCG-3′. The siRNA sequences were designed and synthesized by GenePharma (Shanghai, China). Transfections were carried out according to the instructions of Lipofectamine® 8000 (Beyotime, C0533, China).

## 2.6 RNA extraction, cDNA synthesis, and quantitative real-time PCR

Total RNA was extracted and purified using the RNAiso reagent (Takara, 9109, Japan) according to the Trizol extraction method. 1 μg of total RNA was used as the template for cDNA synthesis, which was performed using the First-Strand cDNA Synthesis Kit (Trans, AT301, China) in a T100™ PCR thermocycler (Bio-Rad, USA). The reverse transcription was carried out in a volume of 20 μl under the following conditions: 42°C for 15 minutes; 85°C for 5 seconds. Quantitative real-time PCR was conducted using the Top Green qPCR Master Mix (Trans, AQ131, China) on an ABI7500 real-time PCR system (Thermo, USA). The reaction was performed in a volume of 20 μl under the following conditions: 94°C for 30 seconds; followed by 40 cycles of 94°C for 5 seconds, and 60°C for 30 seconds. Primers used were as follows: LDHA (forward: 5′-CTCAACCACCTGCTTGTGAACCT-3′, reverse: 5′-AGTGTGCCTGTATGGAGTGGAATG-3′)

and U6 (forward: 5′-CTCGCTTCGGCAGCACA-3′, reverse: 5′-AACGCTTCACGAATTTGC GT-3′). The CT values were used to assess mRNA relative expression levels by the 2-ΔΔCT method, with U6 serving as the internal control. Primers were designed and synthesized by GENEWIZ (Changchun, China), and all samples were run in triplicate.

## 2.7 Western blot

After washing with PBS three times, 120 μL of RIPA lysis buffer containing PMSF (Beyotime, P0013B, China) was added, and cell lysates were collected after centrifugation at 13,300 g for 15 minutes at 4°C. Protein concentration was determined using the BCA Protein Assay Kit (Beyotime, P0012, China). Proteins were separated by SDS-PAGE and then transferred to PVDF membranes in an ice bath at 250 mA for 2 hours. Membranes were blocked with 5% skim milk in TBST at room temperature for 1 hour and washed three times with TBST (10 minutes each) before incubation with the following primary antibodies overnight at 4°C: anti-LDHA (Bioworld, BS6179, USA), anti-AKT (Cell Signaling Technology, 9272S, USA), anti-p-AKT (Ser473) (Cell Signaling Technology, 4060S, USA), and anti-β-Actin (Bioworld, AP0060, USA). The membranes were then washed three times with TBST (10 minutes each) and incubated with secondary antibodies at room temperature for 2 hours. The secondary antibodies used were: HRP-linked Anti-rabbit IgG (Cell Signaling Technology, 7074, USA) and HRP-linked Anti-mouse IgG (Cell Signaling Technology, 7076, USA). Finally, the immunoreactive bands were visualized using ECL substrate (Proteintech, PK10003, USA) and imaged with the Tanon 4600 Automatic Chemiluminescence Imaging System (Tanon, China).

## 2.8 Cell viability assay

Logarithmically growing cells were seeded in 96-well plates at a density of $2–4 \times 10^3$ cells/well and treated with drugs or transfection reagents after settling for 24 hours. After incubation for 24h, 48h, 72h, or 96h, 20 μL of MTT solution (5 mg/mL, pH = 7.4) was added to each well. The incubation was continued for 4 hours before stopping the culture and discarding the supernatant. Each well was then dissolved with 150 μL of DMSO for 10 minutes. The OD value at 492 nm wavelength was measured using a microplate reader. Cell viability was calculated as (OD value of the experimental group—OD value of the blank control group) / (OD value of the control group—OD value of the blank control group) × 100%, with the experiment repeated three times.

## 2.9 Metabolic product and enzyme activity measurement

Logarithmically growing cells were seeded into six-well plates at $3 \times 10^5$ cells per well. After incubation for 24 hours, cell lysates were collected and mixed with assay reagents according to the manufacturer's instructions. The contents of lactate, pyruvate, and ATP were measured using commercial assay kits (Solarbio, BC2235, BC2205, BC0300, China). L-LDH activity was measured using a commercial assay kit (Solarbio, BC0685, China). The content of NAD+ was measured using a commercial assay kit (Beyotime, S0175, China).

## 2.10 Statistical data

Survival rates were analyzed and calculated using Kaplan-Meier and log-rank tests. Quantitative data are expressed as mean ± SEM. Differences between two groups were evaluated using two-tailed Student's t-tests, and one-way ANOVA was used for comparisons among multiple groups. R software was used for statistical analysis and graphical visualization. A p-value < 0.05 was considered statistically significant.

# 3. Results

## 3.1 High levels of LDHA are associated with poor prognosis in LUAD

To validate the role of LDHA in LUAD, the TCGA-LUAD and GSE68571 datasets were analyzed. We discovered that the expression levels of LDHA were significantly higher in tumor tissues compared to normal tissues (Fig 1A, P<0.001). Kaplan-Meier survival analysis showed that higher expression of LDHA was significantly associated with poor overall survival (Fig 1B, P<0.05) and progression-free survival (Fig 1C, P<0.001) in cancer patients. The evaluation of

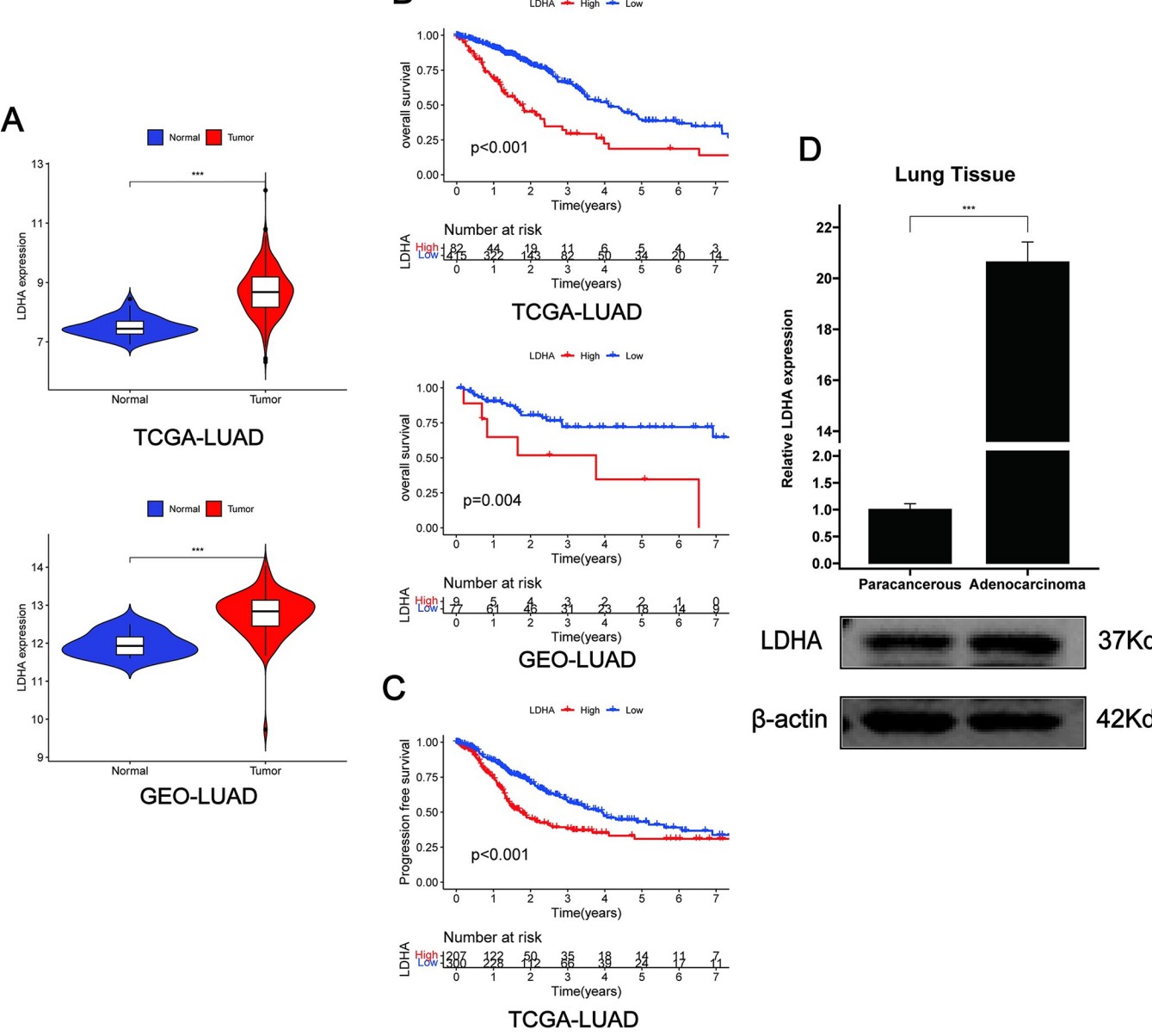

**Fig 1. High expression of LDHA in LUAD is associated with poor prognosis.** (A) Expression of LDHA in cancerous and adjacent non-cancerous samples from the TCGA and GEO datasets. (B) Kaplan-Meier survival curves analyzing the difference in overall survival between high and low LDHA expression groups in cancerous samples from the TCGA and GEO datasets. (C) Kaplan-Meier survival curves analyzing the difference in progression-free survival between high and low LDHA expression groups in cancerous samples from the TCGA dataset. (D) RT-qPCR analysis of LDHA relative mRNA levels and immunoblot experiments of LDHA protein levels in LUAD tissues and paired adjacent non-tumor tissues (n = 3). LUAD: Lung Adenocarcinoma. Data are represented as mean ± SEM. * p < 0.05, ** p < 0.01, *** p < 0.001, ns, not statistically significant.

clinical samples using RT-qPCR found that the expression of LDHA was significantly higher in tumor tissues than in adjacent non-tumor tissues (Fig 1D, P<0.001), which was further confirmed by protein blot analysis (Fig 1D). These results indicate that LDHA is upregulated in LUAD and significantly associated with poor patient prognosis.

To assess the clinical significance of LDHA expression in this disease, we analyzed the relationship between LDHA expression and clinical features. High expression of LDHA was positively correlated with T (P<0.05), N (P<0.05), and stage (P<0.05), but not with age (P = 0.68), sex (P = 0.69), or M (P = 0.17) (S1 Fig). Univariate Cox and multivariate Cox regression analyses of overall survival confirmed that high expression of LDHA is an independent prognostic factor (P<0.05) (S2 Fig), indicating that LDHA has prognostic significance for LUAD patients.

## 3.2 Downregulation of LDHA decreases LUAD cell viability and increases chemotherapy sensitivity to LBP

Given the high expression of LDHA in LUAD, the impact of siRNA-mediated downregulation of LDHA on the viability of LUAD cells and chemotherapy sensitivity to LBP was studied. The expression of LDHA in A549 and NCL-H1975 cells could be effectively reduced by specific siRNAs si-2 and si-3 (Fig 2A), and the activity of LDH in the cells also decreased (Fig 2B). MTT assays were used to analyze the impact of LDHA on the viability of A549 and NCL-H1975 cells and their chemotherapy sensitivity to LBP. Results showed that downregulation of LDHA in A549 and NCL-H1975 cells inhibited cell viability and increased sensitivity to LBP chemotherapy (P<0.05) (Fig 2C and 2D). Moreover, we found that the expression of LDHA in A549 and NCL-H1975 cells increased over time with the action of LBP (Fig 2E). These results suggest that LDHA plays an important role in the viability of A549 and NCL-H1975 cells and their chemotherapy sensitivity to LBP. As lactate dehydrogenase is a key enzyme in the glycolysis pathway, our findings suggest that targeting glycolysis could decrease the viability of A549 and NCL-H1975 cells and increase their chemotherapy sensitivity to LBP.

## 3.3 Oxamate and LBP have synergistic cytotoxic effects on LUAD cells

Oxamate, a pyruvate analog, can directly inhibit the conversion of pyruvate to lactate by LDH, thereby inhibiting cellular glycolysis. We first studied the effect of Oxamate on the viability of A549 and NCL-H1975 cells. Treatment with Oxamate led to inhibition of cell viability (Fig 3A). Subsequently, the effect of Oxamate on LDHA expression and LDH activity was investigated. Results showed that Oxamate could mildly inhibit LDHA expression (Fig 3B) and effectively inhibit LDH activity (Fig 3C). We further studied the effect of combining Oxamate and LBP on the cytotoxicity against A549 and NCL-H1975 cells. Results showed that the combination of Oxamate and LBP was more effective in killing cell viability than either drug alone (Fig 3D). Thus, compared to any single drug treatment, the combination of Oxamate and LBP exerted greater cytotoxic efficacy against LUAD cells. This effect is similar to the results observed with siRNA-mediated LDHA knockdown followed by LBP treatment.

## 3.4 Effect of Oxamate and LBP on partial metabolic products of glycolysis

We had previously demonstrated the close relationship between LDHA and the chemotherapy sensitivity of A549 and NCL-H1975 cells to LBP. To further study the mechanism of Oxamate-induced chemotherapy sensitivity to LBP, the contents of lactate, pyruvate, NAD+, and ATP were measured in A549 and NCL-H1975 cells treated with LBP, Oxamate, or their combination for 24 hours. Results found that treatment with Oxamate for 24 hours led to reduced levels of NAD+, ATP, and lactate in A549 and NCL-H1975 cells. Treatment with LBP for 24 hours led to no significant change in lactate levels in A549 cells, a slight increase in pyruvate,

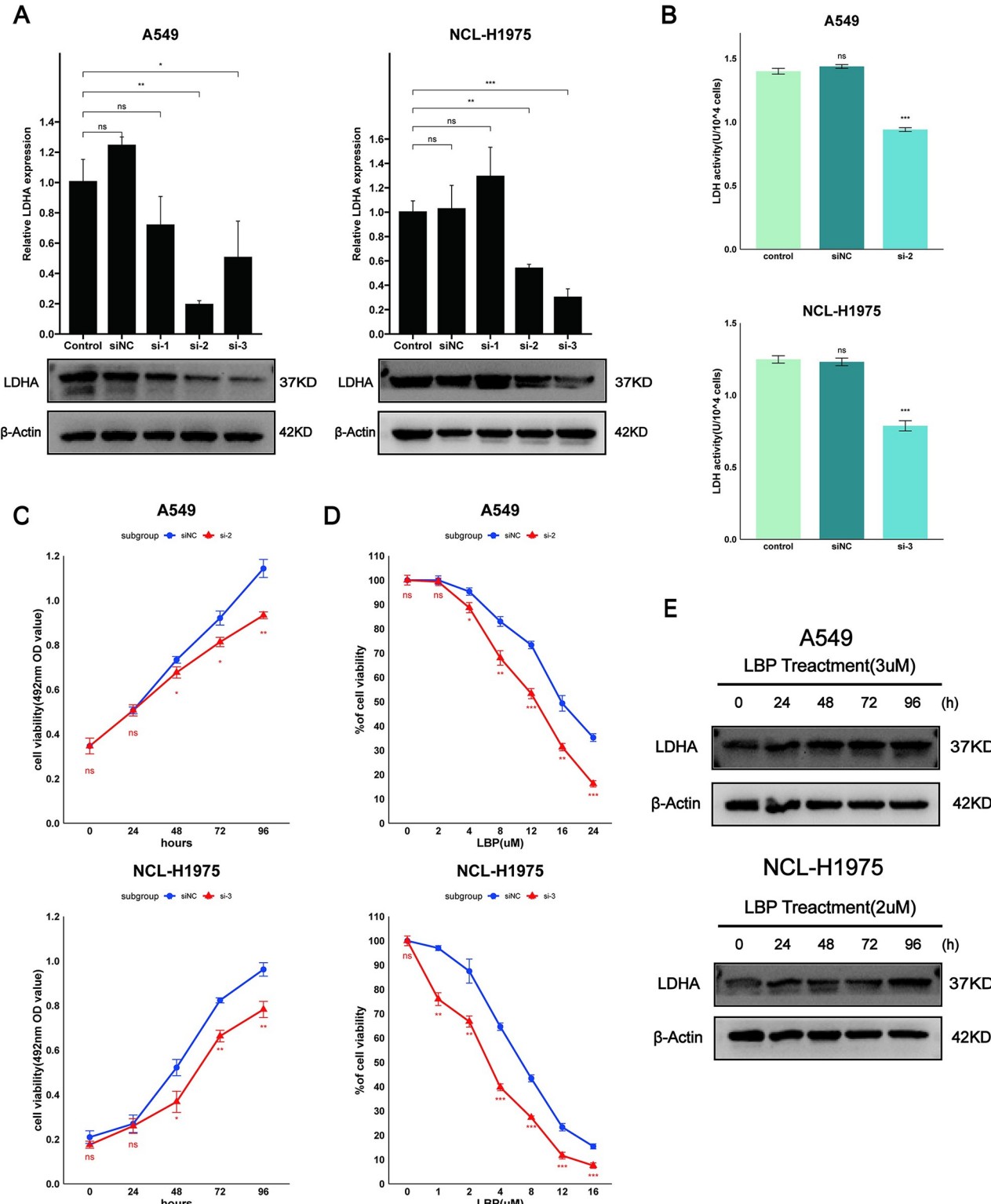

**Fig 2. Knockdown of LDHA reduced cell viability and LDH activity in A549 and NCL-H1975 cells, enhancing chemotherapy sensitivity to LBP.**
(A) LDHA was inhibited in A549 and NCL-H1975 cells. LDHA expression levels were verified by RT-qPCR and protein blot. (B) LDH activity levels were detected from lysates after siRNA transfection in A549 and NCL-H1975 cells. (C) Cell viability of A549 and NCL-H1975 cells after siRNA transfection was analyzed through MTT assay. (D) Chemotherapy sensitivity to LBP in A549 and NCL-H1975 cells after siRNA transfection was analyzed through MTT assay. (E) Changes in LDHA expression levels in A549 and NCL-H1975 cells after treatment with LBP over a time gradient. LBP: Lobaplatin. Data are represented as mean ± SEM. * p < 0.05, ** p < 0.01, *** p < 0.001, ns, not statistically significant.

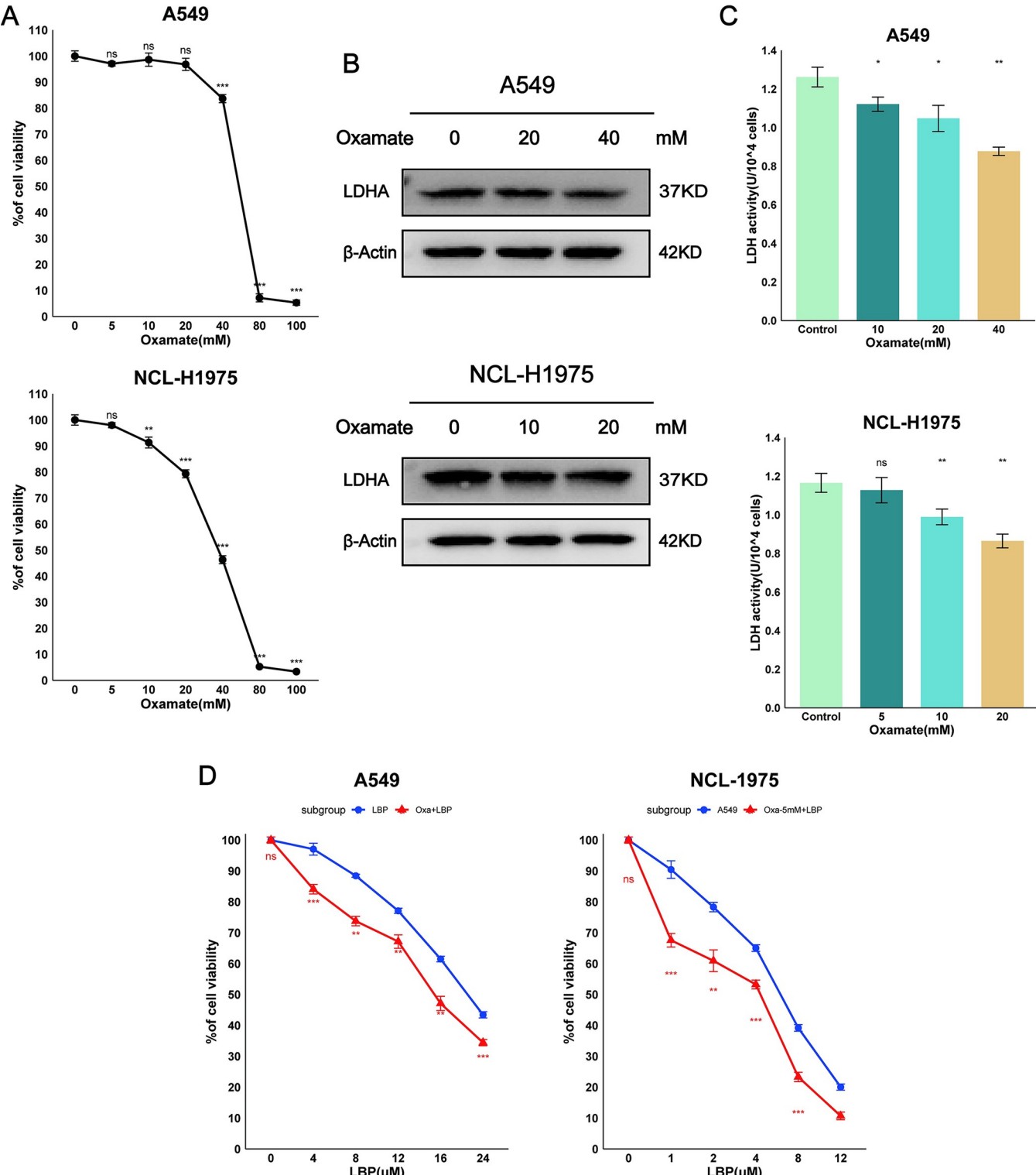

**Fig 3. Oxamate reduces cell viability, LDHA expression, and LDH activity in A549 and NCL-H1975 cells, enhancing the cytotoxic effect of LBP.** (A-C) Changes in cell viability, LDHA expression, and LDH activity in A549 and NCL-H1975 cells after 24 hours of treatment with different concentrations of Oxamate were analyzed through MTT assay, protein blot, and LDH activity test. (D) The cytotoxic effect of combined Oxamate and LBP treatment compared to LBP alone on A549 and NCL-H1975 cells was analyzed through MTT assay. Oxa: Oxamate, LBP: Lobaplatin. Data are represented as mean ± SEM. * $p < 0.05$, ** $p < 0.01$, *** $p < 0.001$, ns, not statistically significant.

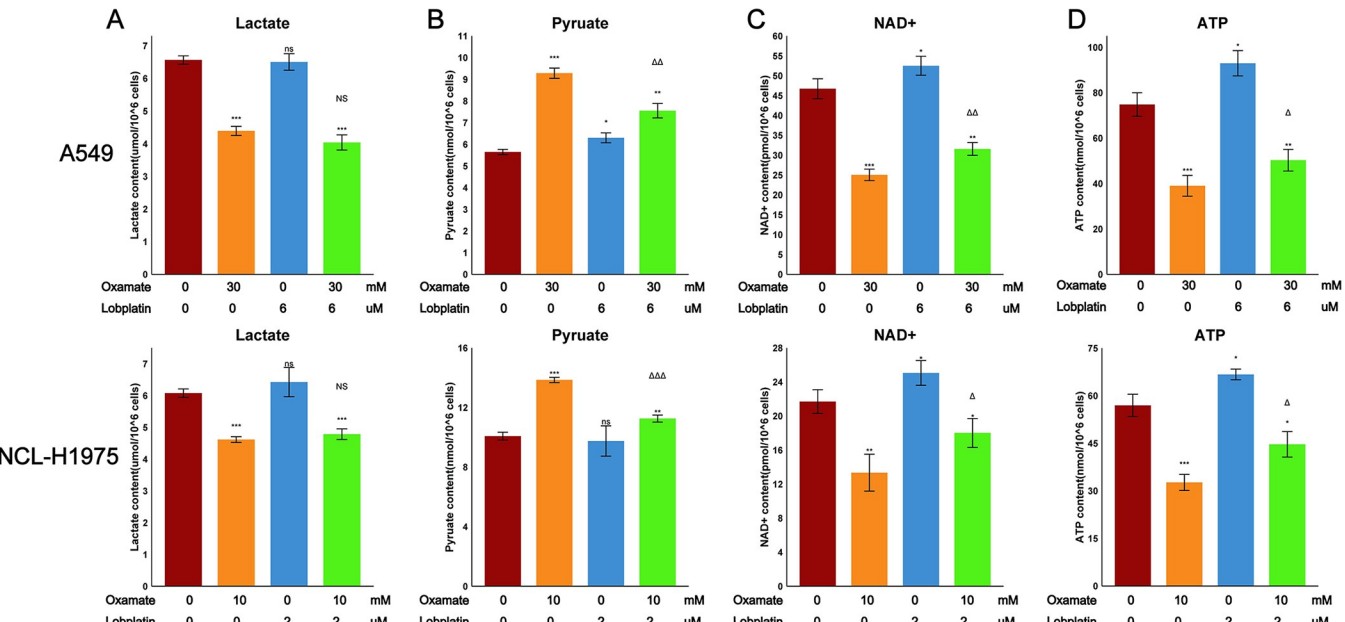

**Fig 4. Effect of Oxamate, LBP, or their combination on partial glycolysis metabolites in A549 and NCL-H1975 cells after 24 hours of treatment.** (A) Lactate. (B) Pyruvate. (C) NAD+. (D) ATP. Data are represented as mean ± SEM. Differences compared to the control group are marked as: * $p < 0.05$, ** $p < 0.01$, *** $p < 0.001$, ns, not statistically significant. Differences compared to the Oxamate (Oxa) group are marked as: Δ $p < 0.05$, ΔΔ $p < 0.01$, ΔΔΔ $p < 0.001$, NS, not statistically significant.

NAD+, and ATP levels, and no significant change in pyruvate and lactate levels in NCL-H1975 cells, with a slight increase in NAD+ and ATP levels. However, the combination of Oxamate and LBP partially mitigated the metabolic effects of Oxamate on metabolic products other than lactate (Fig 4A–4D).

## 3.5 Supplementing NAD+ mitigates the metabolic inhibition and increased chemotherapy sensitivity induced by Oxamate

The NADH cycle plays an important role in glycolysis. NADH provides hydrogen ions when pyruvate is converted to lactate, and NAD+ accepts hydrogen ions released from glyceraldehyde-3-phosphate, ensuring the forward progression of glycolysis (Fig 5A). Moreover, we had previously demonstrated the crucial role of LDHA in regulating the glycolysis and survival capabilities of A549 and NCL-H1975 cells. The decrease in expression and activity of LDHA was shown to lead to glycolysis blockade and reduced ATP. By supplementing NAD+, we facilitated the normal progression of glycolysis. Observing the changes in glycolysis and survival capabilities of A549 and NCL-H1975 cells after supplementing NAD+ revealed that NAD + supplementation could alleviate the inhibition of glycolysis by Oxamate to some extent (Fig 5B–5D). Furthermore, NAD+ also alleviated the cytotoxicity of the combination of Oxamate and LBP (Fig 5E). These results further support the notion that Oxamate increases chemotherapy sensitivity to LBP by inhibiting cellular glycolysis.

## 3.6 LDHA inhibits glycolysis-produced ATP, thereby reducing AKT phosphorylation and increasing chemotherapy sensitivity to LBP

To investigate the biological pathways involved by LDHA in LUAD, we performed KEGG enrichment analysis on tumor samples included in the TCGA dataset. S1 Table lists the gene

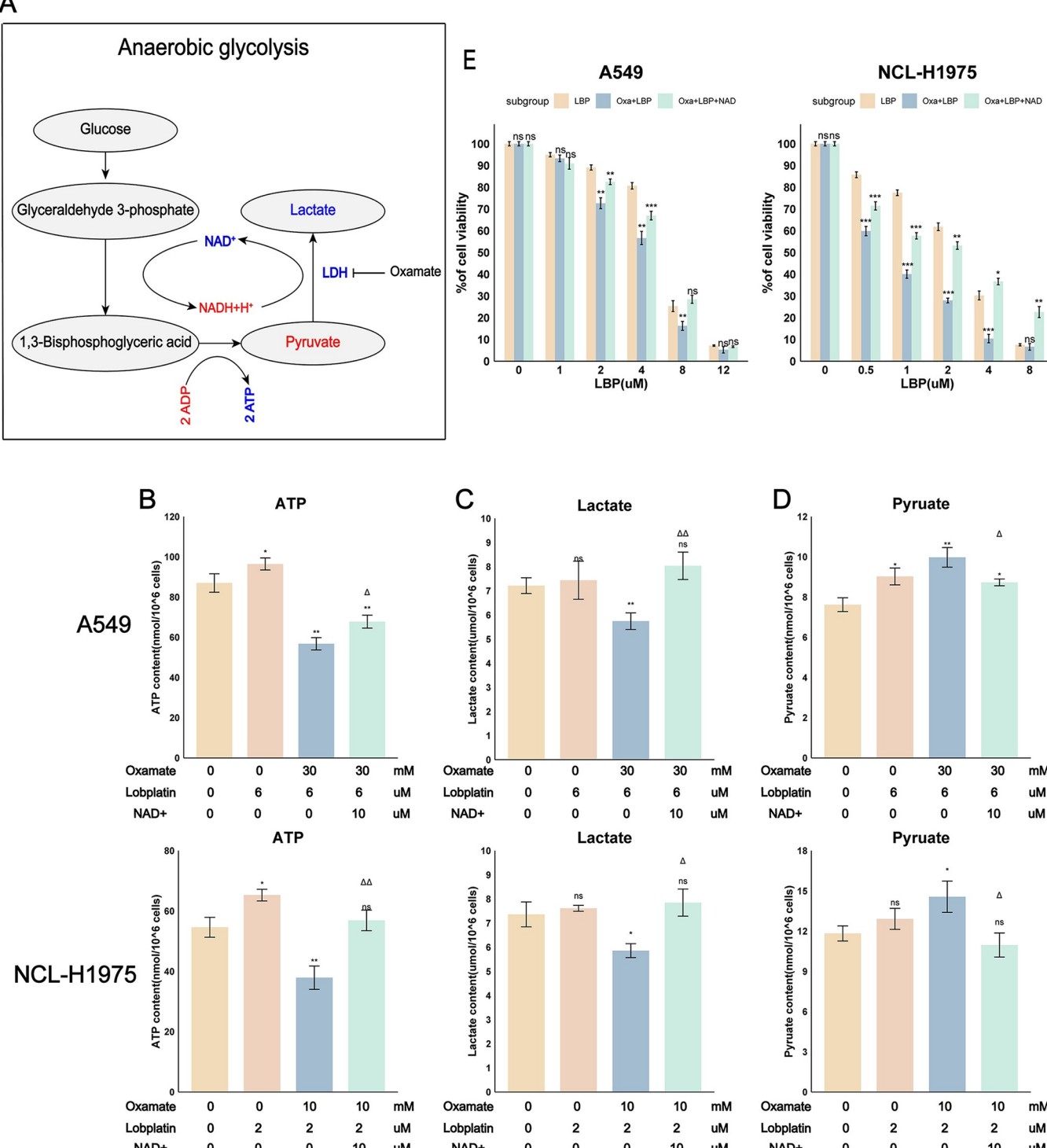

**Fig 5. Supplementing NAD+ counters the effects of Oxamate on metabolism and chemotherapy sensitivity.** (A) Schematic of Oxamate's effect on anaerobic glycolysis. Blue indicates downregulation, red indicates an increase. (B-D) Changes in ATP, lactate, and pyruvate levels in A549 and NCL-H1975 cells after the combination of Oxamate and LBP with the addition of NAD+. (E) The impact of the aforementioned conditions on cell viability in A549 and NCL-H1975 cells was analyzed through MTT assay. Oxa: Oxamate, LBP: Lobaplatin. Data are represented as mean ± SEM. Differences compared to the control group are marked as: * p < 0.05, ** p < 0.01, *** p < 0.001, ns, not statistically significant. Differences compared to the Oxamate combined with LBP group are marked as: Δp < 0.05, ΔΔ p < 0.01, ΔΔΔ p < 0.001, NS, not statistically significant.

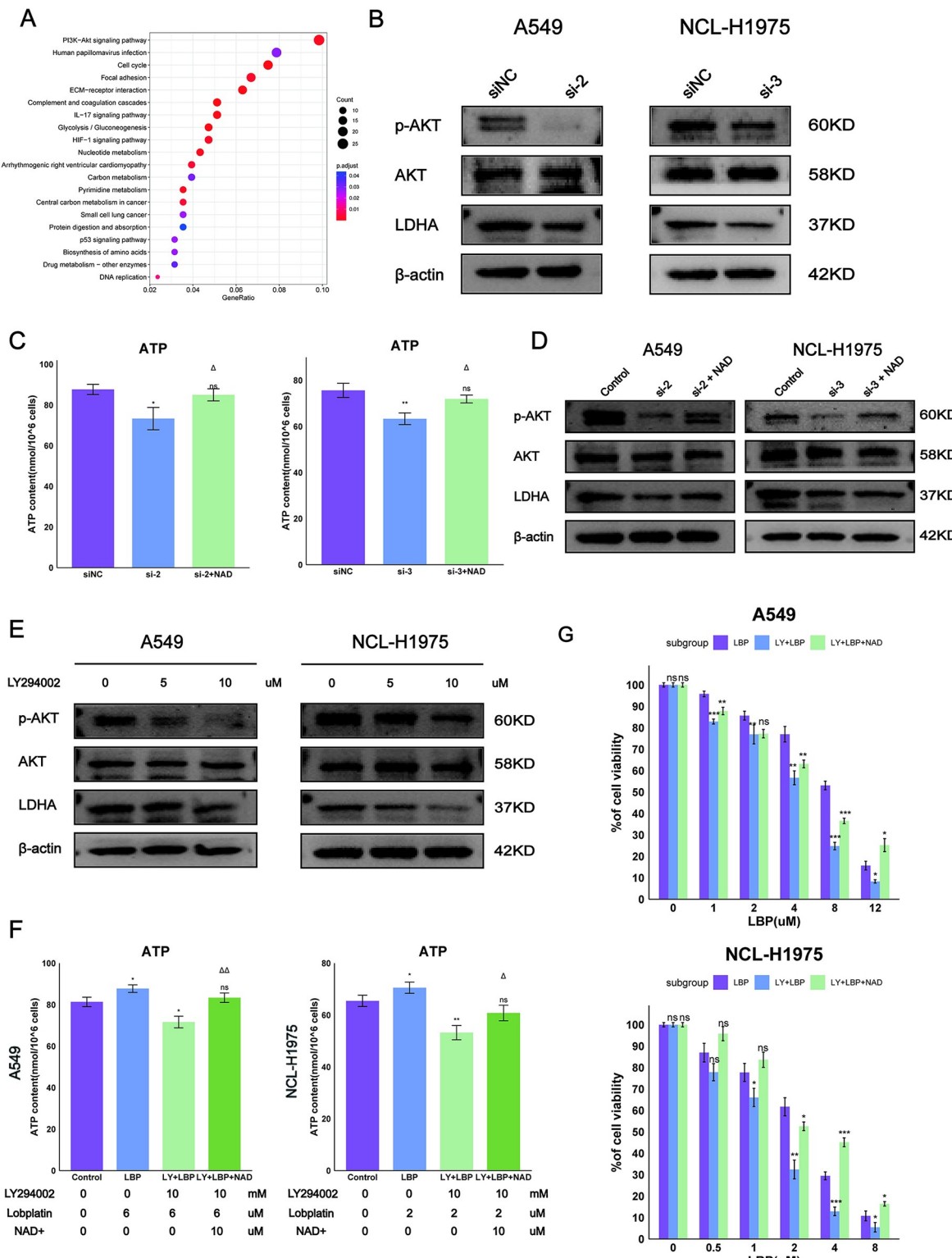

**Fig 6. Inhibition of LDHA leads to decreased ATP, reduced AKT phosphorylation, and enhanced chemotherapy sensitivity to LBP in A549 and NCL-H1975 cells.** (A) KEGG enrichment analysis for the high LDHA expression group. Blue indicates downregulation, red indicates an increase. (B) The effect of LDHA inhibition on p-Akt and Akt protein expression in A549 and NCL-H1975 cells. (C-D) The impact of LDHA inhibition followed by NAD+ supplementation on ATP content and p-Akt and Akt protein expression in A549 and NCL-H1975 cells. (E) The effect of LY294002 inhibition of p-Akt phosphorylation on LDHA protein expression in A549 and NCL-H1975

cells. (F) Changes in ATP content in A549 and NCL-H1975 cells after the combination of LY294002 and Lobaplatin with NAD
+ supplementation. (G) The impact of the aforementioned conditions (F) on cell viability in A549 and NCL-H1975 cells was analyzed
through MTT assay. LY: LY294002, LBP: Lobaplatin. Data are represented as mean ± SEM. Differences compared to the control group are
marked as: * $p < 0.05$, ** $p < 0.01$, *** $p < 0.001$, ns, not statistically significant. Differences compared to the LY combined with LBP
group are marked as: $\Delta p < 0.05$, $\Delta\Delta p < 0.01$, $\Delta\Delta\Delta p < 0.001$, NS, not statistically significant.

pathways enriched in the high LDHA expression group. We found that genes in the high
LDHA expression group were mainly enriched in the PI3K/AKT signaling pathway (Fig 6A).
It has been reported that the lack of LDHA inhibits ATP production related to glycolysis, lead-
ing to a decrease in PI3K-dependent activation of Akt kinase [13]. Further protein blot analysis
confirmed that inhibition of LDHA led to downregulation of P-AKT, but had no effect on
total AKT expression (Fig 6B). We then examined ATP production and found that inhibition
of LDHA reduced ATP content, which could be partially mitigated by adding NAD+ (Fig 6C).
In the condition of LDHA inhibition, adding NAD+ increased the expression of P-AKT pro-
tein (Fig 6D), indicating that LDHA promotes the NADH cycle to produce ATP, ultimately
enhancing P-AKT levels. Therefore, the AKT pathway may be a downstream effector related
to LDHA in cancer.

Additionally, adding LY294002 to inhibit the PI3K-Akt pathway led to reduced LDHA pro-
tein expression (Fig 6E), and we then checked ATP content. Results showed that LY294002
could inhibit cellular ATP production, which could be partially mitigated by adding NAD+
(Fig 6F). Meanwhile, MTT analysis of the combined use of LY294002 and LBP found that
LY294002 also enhanced the chemotherapy sensitivity of A549 and NCL-H1975 cells to LBP,
and the cytotoxicity could be partially mitigated by NAD+ (Fig 6G). These results indicate that
the PI3K-Akt pathway also regulates LDHA expression, affecting cellular ATP production and
chemotherapy sensitivity to LBP.

## 4. Discussion

Lung adenocarcinoma (LUAD), as one of the most challenging subtypes of non-small cell lung
cancer (NSCLC) in clinical practice, presents treatment difficulties primarily due to the
tumor's high resistance to conventional chemotherapy drugs. In recent years, with a deeper
understanding of the metabolic characteristics of cancer cells, LDHA, a key enzyme in the gly-
colysis process, has garnered widespread attention from researchers. In the tumor microenvi-
ronment, cancer cells primarily utilize glucose to produce energy through glycolysis, even
under normoxic conditions, a phenomenon known as the "Warburg effect" [9]. The Warburg
effect is one of the main hallmarks of cancer cells. LDHA is the major subunit of LDH, a key
enzyme involved in the Warburg effect [14]. Previous studies have shown that LDHA expres-
sion is elevated in most types of cancer cells [15–17]. Our study, through bioinformatics analy-
sis and clinical sample testing, found that LDHA expression in LUAD tissues is significantly
higher than in normal tissues and that its high expression is closely related to poor patient
prognosis. This result has been validated in our previous construction of a prognostic model
for LUAD [18].

In the glycolysis process, LDHA converts pyruvate to lactate and energy, simultaneously
regenerating NADH to NAD+, playing a crucial role in maintaining the cell's redox homeosta-
sis [19]. By consuming preserved pyruvate and regenerating NAD+, LDHA maintains the
NAD+/NADH redox balance necessary for sufficient glycolysis [20]. While the mitochondrial
electron transport chain (ETC) can regenerate NAD+ from NADH, this process requires the
transport of electrons into the mitochondrial matrix via cytoplasmic shuttles such as the
malate-aspartate shuttle. This shuttle involves multiple steps and is kinetically slower than the

direct conversion of pyruvate to lactate [20], especially when LDHA is abundant [21, 22]. Our study demonstrates that LDHA facilitates glycolysis by promoting the NADH/NAD+ cycle, highlighting its critical role in ensuring efficient energy production in cancer cells.

Like other platinum drugs, LBP possesses anticancer activity but has unique advantages in chemotherapy, with no significant nephrotoxicity or neurotoxicity [23]. However, as treatment progresses, tumor cells often gradually develop resistance to LBP, limiting its long-term therapeutic effect. We found that LBP treatment led to an increase in LDHA expression in LUAD cells, resulting from the induction of LDHA mRNA expression. This adaptive response highlights the potential for targeting glycolysis to enhance the effectiveness of chemotherapy and prevent the development of drug resistance. Targeted inhibition of LDHA, through either siRNA or the small molecule inhibitor Oxamate, resulted in a significant reduction in glycolytic activity and cell viability in LUAD cells. More importantly, the combination of LDHA inhibition with LBP treatment produced a synergistic antitumor effect, indicating that disrupting glycolysis can enhance the cytotoxic efficacy of LBP.

This synergy suggests that LBP treatment triggers a feedback mechanism where increased LDHA expression helps cancer cells survive under the stress of chemotherapy by maintaining their glycolytic pathway, which is crucial for energy production. Targeting LDHA with LDHA siRNA or Oxamate disrupts this feedforward cycle, increasing LBP chemotherapy sensitivity. The enhanced chemotherapy sensitivity observed with LDHA inhibition underscores the potential of metabolic targeting as a strategy to overcome drug resistance. By disrupting the glycolytic pathway, which cancer cells heavily rely on for energy and biosynthetic precursors, we effectively reduce the cells' ability to survive and proliferate under chemotherapeutic stress. This finding provides new insights into overcoming chemotherapy resistance, echoing previous research [24–27] pointing to the potential of targeting the glycolysis pathway to enhance tumor chemotherapy sensitivity.

The PI3K/AKT signaling pathway is involved in regulating a variety of cellular activities, including cell growth, migration, differentiation, apoptosis, and energy metabolism [28–30]. Studies have confirmed that AKT increases the level of glycolysis in tumor cells without affecting aerobic oxidation, providing ample materials for biosynthesis and promoting total ATP production [30]. Abnormal activation of the PI3K/AKT signaling cascade occurs in various tumors [31, 32]. Our study found that in the high LDHA expression group, genes were mainly enriched in the PI3K/AKT signaling pathway. Subsequent siRNA targeting LDHA experiments revealed that downregulating LDHA was accompanied by a decrease in ATP and p-AKT. Experiments targeting AKT phosphorylation with LY294002 found that blocking the PI3K/AKT signaling pathway was also accompanied by a decrease in ATP and enhanced antitumor effect of LBP, with a corresponding decrease in LDHA expression. ATP appears to link the PI3K/AKT signaling pathway and LDHA in a feedback loop, playing an important role in cancer cells' chemotherapy sensitivity to LBP.

Despite producing less ATP per glucose molecule compared to mitochondrial oxidative phosphorylation, glycolysis offers several key advantages that are particularly beneficial to rapidly proliferating cancer cells [33]. Firstly, the rate of glycolysis and the conversion of glucose to lactate accelerates, leading to faster and more ATP production. The ATP production pathway of glycolysis, being high-rate but low-yield, has a selective advantage in the competition for shared energy resources [34]. The ATP production speed of glycolysis may be up to 100 times faster than oxidative phosphorylation [35]. However, the low yield of ATP from glycolysis is sufficient to meet the intracellular demands. Secondly, besides ATP, cancer cells require further metabolic intermediates and precursors essential for the biosynthesis of macromolecules, indispensable final components for increasing tumor mass during growth and proliferation processes [36, 37]. ATP is not only a source of energy for cellular life activities but also

plays a central role in the proliferation, migration, and resistance formation of tumor cells [38]. Studies indicate that tumor cells enhance ATP production through glycolysis to meet the energy demands of rapid growth, thereby promoting tumor development and progression [34]. Therefore, targeting glycolysis to disrupt ATP production and reduce the availability of biosynthetic intermediates presents a promising strategy for cancer therapy [39]. By inhibiting glycolytic enzymes such as LDHA, it is possible to impede the metabolic flexibility of cancer cells, making them more susceptible to chemotherapy and reducing their ability to proliferate and metastasize. This approach not only hampers the energy production of cancer cells but also interferes with their anabolic processes, providing a multifaceted avenue to counteract tumor growth and resistance mechanisms.

In summary, our study highlights the pivotal role of LDHA in maintaining energy metabolism and chemotherapy sensitivity in lung adenocarcinoma. Targeting LDHA disrupts glycolysis, reduces ATP production, and enhances the efficacy of Lobaplatin, offering a promising approach to overcome drug resistance in LUAD. This insight lays the groundwork for new treatment strategies that exploit metabolic vulnerabilities to improve cancer therapy outcomes.

## Supporting information

**S1 Fig. Analysis of the correlation between LDHA expression and clinical characteristics.** (A) T1+T2 vs. T3+T4. (B) N0 vs. ≥N1 (N1+N2). (C) Comparison among four levels. (D) Under 60 years old vs. ≥60 years old. (E) Female vs. male. (F) M0 vs. M1.
(TIF)

**S2 Fig. LDHA expression is an independent prognostic factor for the overall survival of LUAD patients.** (A) Univariate Cox regression analysis of overall survival with LDHA expression and clinical characteristics. (B) Multivariate Cox regression analysis of overall survival with LDHA expression and clinical characteristics.
(TIF)

**S1 Table. KEGG enrichment results of differentially expressed genes between low and low LDHA expression groups.**
(DOCX)

**S1 Raw images.**
(PDF)

## Author Contributions

**Data curation:** Xuguang Mi, Junjie Hou.

**Formal analysis:** Siyu Yuan, Wenjie Ou.

**Methodology:** Siyu Yuan, Wenjie Ou, Xuguang Mi.

**Resources:** Junjie Hou.

**Software:** Wenjie Ou.

**Visualization:** Siyu Yuan.

**Writing – original draft:** Siyu Yuan, Wenjie Ou.

**Writing – review & editing:** Wenjie Ou, Xuguang Mi, Junjie Hou.

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
