## [Decision Letter · Decision Letter 0]

5 Aug 2024

PONE-D-24-25075Enhancing Lobaplatin Sensitivity in Lung Adenocarcinoma through inhibiting LDHA-Targeted Metabolic PathwaysPLOS ONE

Dear Dr. Hou,

Thank you for submitting your manuscript to PLOS ONE and thank you for the patience while we wait for the reviewers' comments, which normally takes longer in the summer time. Your manuscript has now been assessed by two independent referees. While both of them expressed interest in the study, there are some concerns that should be addressed before we can move forward, including but not limited to lack of control groups and clinical relevance. Therefore, we invite you to submit a revised version of the manuscript that addresses the points raised during the review process. Please refer to the detailed comments attached below.

We look forward to receiving your revised manuscript.

Kind regards,

Zhiming Li, Ph.D.

Academic Editor

PLOS ONE

2. PLOS requires an ORCID iD for the corresponding author in Editorial Manager on papers submitted after December 6th, 2016. Please ensure that you have an ORCID iD and that it is validated in Editorial Manager. To do this, go to ‘Update my Information’ (in the upper left-hand corner of the main menu), and click on the Fetch/Validate link next to the ORCID field. This will take you to the ORCID site and allow you to create a new iD or authenticate a pre-existing iD in Editorial Manager. Please see the following video for instructions on linking an ORCID iD to your Editorial Manager account: https://www.youtube.com/watch?v=_xcclfuvtxQ".

 [Jilin Province Health Commission Project(2021LC058)

Jilin Province Science and Technology Development Plan Project(20230402007GH)

Jilin Province health science and technology ability improvement project(2023LC043)].  

7. Please upload a copy of  Supplementary Table S1,  Supplementary Figure S1 and Supplementary Figure S2

 to which you refer in your text on page 22. Please amend the file type to 'Supporting Information'. If the Supplementary file is no longer to be included as part of the submission please remove all reference to it within the text.

9. We notice that your supplementary table is included in the manuscript file. Please remove them and upload them with the file type 'Supporting Information'. Please ensure that each Supporting Information file has a legend listed in the manuscript after the references list.

Reviewers' comments:

Reviewer's Responses to Questions

**Comments to the Author**

1. Is the manuscript technically sound, and do the data support the conclusions?

Reviewer #1: Partly

Reviewer #2: Partly

2. Has the statistical analysis been performed appropriately and rigorously? 

Reviewer #1: Yes

Reviewer #2: I Don't Know

3. Have the authors made all data underlying the findings in their manuscript fully available?

Reviewer #1: Yes

Reviewer #2: Yes

4. Is the manuscript presented in an intelligible fashion and written in standard English?

Reviewer #1: Yes

Reviewer #2: Yes

5. Review Comments to the Author

Reviewer #1: This study analyzed public databases and found that LDHA was significantly higher in tumor tissues, a conclusion that was validated in 3 pairs of tumor and adjacent tissues. Furthermore, through in vitro cell experiments, it was found that downregulation of LDHA reduced the viability of LUAD cells and increased sensitivity to LBP chemotherapy. The combination of LDHA inhibitors and oxaliplatin produced a synergistic anti-tumor effect. However, this article has several shortcomings, including but not limited to:

1. LBP is not a commonly used chemotherapy drug for lung adenocarcinoma in clinical practice, while cisplatin or carboplatin are more commonly used. Further experiments are needed to verify if similar results can be obtained with these drugs.

2. The expression of LDHA in tumor tissues in this study mainly comes from public databases, while the sample size verified by this research institution is only 3 cases, making the results unreliable.

3. The study on LDHA sensitizing LBP is limited to cell experiments and has not been validated in animal models, leading to a certain degree of uncertainty in the conclusions.

Reviewer #2: This is an interesting study on the role of Lactate pathway in the development of drug resistance in LUAD. The energy supply for cancer cells is highly dependent on the Lactose pathway and DNA repair and Nucleotide synthesis are highly energy consuming processes. Therefore it is not surprising, that LDHA inhibition enhances chemosensitivity and reduces drug resistance. The study is so far nicely conducted, but the main limitation is, that it is lacking control groups to evaluate the specific effect of LDHA inhibition on drug resistance and distinguish it from an unspecific energy depleting mechanism. For further review of this work, the introduction of control groups are mandatory.

6. PLOS authors have the option to publish the peer review history of their article (what does this mean?). If published, this will include your full peer review and any attached files.

Reviewer #1: No

Reviewer #2: No

---

## [Author Response · Author response to Decision Letter 0]

9 Aug 2024

Dear editor and reviewers:

We sincerely appreciate the constructive feedback provided by you and the reviewers on our manuscript entitled "Enhancing Lobaplatin Sensitivity in Lung Adenocarcinoma through inhibiting LDHA-Targeted Metabolic Pathways" (PONE-D-24-25075). We have carefully considered all the comments and have made the necessary revisions to address them. Please find below a detailed point-by-point response to the reviewers’ comments, along with the corresponding changes made to the manuscript. The revised portions are marked in red in the revised manuscript with track changes. We thank Dr. Zhiming Li and reviewers for editing this manuscript.

We believe that the quality of the revised manuscript has been greatly improved with the help of you and the reviewers. We hope that the revised manuscript will be satisfactory and acceptable for publication in PLOS ONE. The principal amendments to the paper and responses to the reviewers' comments are as follows:

Responses to the Editor’s Comments

1.Please ensure that your manuscript meets PLOS ONE's style requirements, including those for file naming.

Response: We have ensured that the revised manuscript meets PLOS ONE's style requirements, including those for file naming.

2.PLOS requires an ORCID iD for the corresponding author in Editorial Manager.

Response: We have ensured that the corresponding author has an ORCID iD and it is validated in Editorial Manager.

3.Thank you for stating the following financial disclosure: "Jilin Province Health Commission Project(2021LC058) Jilin Province Science and Technology Development Plan Project(20230402007GH) Jilin Province health science and technology ability improvement project(2023LC043)." Please state what role the funders took in the study.

Response: Financial support was provided by Jilin Province Health Commission Project (2021LC058), Jilin Province Science and Technology Development Plan Project (20230402007GH), and Jilin Province Health Science and Technology Ability Improvement Project (2023LC043). The funders had no role in study design, data collection and analysis, decision to publish, or preparation of the manuscript.

4.Data Availability Statement confirmation.

Response: We confirm that our submission contains all raw data required to replicate the results of our study. All relevant data are within the manuscript and its Supporting Information files.

5.Your ethics statement should only appear in the Methods section of your manuscript.

Response: We have ensured that the ethics statement is included only in the Methods section of the manuscript.

6.Provide the original uncropped and unadjusted images underlying all blot or gel results.

Response: We have provided the original uncropped and unadjusted images underlying all Western blot results as required. These images have been compiled into a PDF file named "S1_raw_images" and uploaded as Supporting Information.

7.Upload Supplementary Table S1, Supplementary Figure S1, and Supplementary Figure S2 as 'Supporting Information'.

Response: We have uploaded the supplementary files as 'Supporting Information' and included captions for these files at the end of the manuscript.

8.Include captions for your Supporting Information files at the end of your manuscript.

Response: We have included captions for all Supporting Information files at the end of the manuscript and updated the in-text citations accordingly.

9.Remove supplementary table from the manuscript file and upload it with the file type 'Supporting Information'.

Response: We have removed the supplementary table from the manuscript file and uploaded it as 'Supporting Information'.

Responses to the Reviewer’s Comments

Reviewer #1:

1.Comment: LBP is not a commonly used chemotherapy drug for lung adenocarcinoma in clinical practice, while cisplatin or carboplatin are more commonly used. Further experiments are needed to verify if similar results can be obtained with these drugs.

Response: In 2021, lobaplatin was incorporated into the latest edition of the "Chinese Medical Association Lung Cancer Clinical Diagnosis and Treatment Guidelines" as a first-line treatment for NSCLC patients[ [Oncology Society of Chinese Medical Association guideline for clinical diagnosis and treatment of lung cancer (2021 edition)]. Zhonghua zhong liu za zhi [Chinese journal of oncology]. 2021;43. DOI: 10.3760/cma.j.cn112137-20210207-00377]. Given this endorsement and the prevalence of lobaplatin-based chemotherapy regimens in our department, we chose to focus on lobaplatin rather than cisplatin or carboplatin for this study. These guidelines and clinical practices support our decision to utilize lobaplatin.

2.Comment: The expression of LDHA in tumor tissues in this study mainly comes from public databases, while the sample size verified by this research institution is only 3 cases, making the results unreliable.

Response: We appreciate your concern regarding the sample size. We utilized data from TCGA and GEO databases, which include several hundred LUAD tissue samples and their matched adjacent non-cancerous tissues, showing consistent overexpression of LDHA in LUAD. To further validate these findings, we conducted additional experiments at Jilin Province People's Hospital, analyzing 3 pairs of LUAD tissues and adjacent non-cancerous tissues. Due to practical constraints, including the limited timeframe of a master's study, the complexities of tissue collection, patient availability, and reduced patient numbers during the COVID-19 pandemic, we were able to include 3 samples. 

3.Comment: The study on LDHA sensitizing LBP is limited to cell experiments and has not been validated in animal models, leading to a certain degree of uncertainty in the conclusions.

Response: Due to laboratory constraints and time limitations, we were unable to conduct animal model experiments to validate our findings. We acknowledge this limitation and suggest that future studies should include animal models for empirical validation to strengthen the conclusions.

Reviewer #2:

1.Comment: This is an interesting study on the role of Lactate pathway in the development of drug resistance in LUAD. The energy supply for cancer cells is highly dependent on the Lactose pathway and DNA repair and Nucleotide synthesis are highly energy consuming processes. Therefore it is not surprising, that LDHA inhibition enhances chemosensitivity and reduces drug resistance. The study is so far nicely conducted, but the main limitation is, that it is lacking control groups to evaluate the specific effect of LDHA inhibition on drug resistance and distinguish it from an unspecific energy depleting mechanism. For further review of this work, the introduction of control groups are mandatory.

Response: In our study, our primary focus was to enhance the chemotherapy sensitivity of LUAD cells to Lobaplatin (LBP) through LDHA inhibition. We acknowledge the importance of distinguishing the specific effects of LDHA inhibition from broader metabolic disruptions and energy depletion mechanisms. To address these concerns and validate our findings, future studies will include the use of energy depletion controls, such as 2-deoxy-D-glucose (2-DG), a known glycolysis inhibitor, to compare the effects of general glycolytic inhibition with those of specific LDHA inhibition. Additionally, we will incorporate metabolic inhibitors that do not directly target glycolysis to help clarify whether the observed effects are due to specific LDHA inhibition or broader metabolic disruptions. We appreciate your guidance on this matter and understand the importance of these control experiments for validating our findings and addressing the limitations of the current study. Your feedback has been invaluable in improving the rigor and robustness of our research. Thank you for your insightful comments.

Sincerely,

Junjie Hou

Department of Comprehensive Oncology, Jilin Provincial People's Hospital, 1183 Gongnong Street, Changchun 130021, Jilin, China.

E-mail: houjunjie1979@163.com

---

## [Decision Letter · Decision Letter 1]

8 Sep 2024

Enhancing lobaplatin sensitivity in lung adenocarcinoma through inhibiting LDHA-targeted metabolic pathways

PONE-D-24-25075R1

Dear Dr. Hou,

Thank you for the efforts in revising the manuscript. While one of the reviewers still has reservations about the manuscript, the editorial team believes that the conclusions can be supported by the presented evidence. Therefore, we’re pleased to inform you that your manuscript has been judged scientifically suitable for publication and will be formally accepted for publication once it meets all outstanding technical requirements.

Kind regards,

Zhiming Li, Ph.D.

Academic Editor

PLOS ONE

Additional Editor Comments:

Both reviewers have agreed that the conclusions are supported by the presented evidence in the manuscript. Therefore, while addressing reviewer #2's comments to include the control groups can further improve the validity of the conclusions, I don't think it's essential at this stage.

Reviewers' comments:

Reviewer's Responses to Questions

**Comments to the Author**

1. If the authors have adequately addressed your comments raised in a previous round of review and you feel that this manuscript is now acceptable for publication, you may indicate that here to bypass the “Comments to the Author” section, enter your conflict of interest statement in the “Confidential to Editor” section, and submit your "Accept" recommendation.

Reviewer #1: All comments have been addressed

Reviewer #2: All comments have been addressed

2. Is the manuscript technically sound, and do the data support the conclusions?

Reviewer #1: Yes

Reviewer #2: Yes

3. Has the statistical analysis been performed appropriately and rigorously? 

Reviewer #1: Yes

Reviewer #2: I Don't Know

4. Have the authors made all data underlying the findings in their manuscript fully available?

Reviewer #1: Yes

Reviewer #2: No

5. Is the manuscript presented in an intelligible fashion and written in standard English?

Reviewer #1: Yes

Reviewer #2: Yes

6. Review Comments to the Author

Reviewer #1: (No Response)

Reviewer #2: The critical point concerning the control groups has unfortunately not been addessed. I rhink without those controls the conclusion outlined in the title can not been drawn from this data conclusively.

7. PLOS authors have the option to publish the peer review history of their article (what does this mean?). If published, this will include your full peer review and any attached files.

Reviewer #1: No

Reviewer #2: No

---

## [Editor Report · Acceptance letter]

13 Sep 2024

PONE-D-24-25075R1 

PLOS ONE

Dear Dr. Hou, 

I'm pleased to inform you that your manuscript has been deemed suitable for publication in PLOS ONE. Congratulations! Your manuscript is now being handed over to our production team.

Kind regards, 

on behalf of

Dr. Zhiming Li 

Academic Editor

PLOS ONE